# Current Understanding of the Relationship of HDL Composition, Structure and Function to Their Cardioprotective Properties in Chronic Kidney Disease

**DOI:** 10.3390/biom10091348

**Published:** 2020-09-21

**Authors:** Gunther Marsche, Gunnar H. Heine, Julia T. Stadler, Michael Holzer

**Affiliations:** 1Otto Loewi Research Center, Division of Pharmacology, Medical University of Graz, 8010 Graz, Austria; julia.stadler@medunigraz.at (J.T.S.); michael.holzer@medunigraz.at (M.H.); 2Agaplesion Markus Krankenhaus, 60431 Frankfurt, Germany; Gunnar.Heine@uks.eu; 3Faculty of Medicine, Saarland University, D-66424 Homburg, Germany

**Keywords:** HDL proteome, HDL cholesterol efflux capacity, kidney failure

## Abstract

In the general population, the ability of high-density lipoproteins (HDLs) to promote cholesterol efflux is a predictor of cardiovascular events, independently of HDL cholesterol levels. Although patients with chronic kidney disease (CKD) have a high burden of cardiovascular morbidity and mortality, neither serum levels of HDL cholesterol, nor cholesterol efflux capacity associate with cardiovascular events. Important for the following discussion on the role of HDL in CKD is the notion that traditional atherosclerotic cardiovascular risk factors only partially account for this increased incidence of cardiovascular disease in CKD. As a potential explanation, across the spectrum of cardiovascular disease, the relative contribution of atherosclerotic cardiovascular disease becomes less important with advanced CKD. Impaired renal function directly affects the metabolism, composition and functionality of HDL particles. HDLs themselves are a heterogeneous population of particles with distinct sizes and protein composition, all of them affecting the functionality of HDL. Therefore, a more specific approach investigating the functional and compositional features of HDL subclasses might be a valuable strategy to decipher the potential link between HDL, cardiovascular disease and CKD. This review summarizes the current understanding of the relationship of HDL composition, metabolism and function to their cardio-protective properties in CKD, with a focus on CKD-induced changes in the HDL proteome and reverse cholesterol transport capacity. We also will highlight the gaps in the current knowledge regarding important aspects of HDL biology.

## 1. Introduction

High-density lipoprotein (HDL) cholesterol (HDL-C) is inversely associated with the risk of atherosclerotic cardiovascular disease (ASCVD) and is a key component in predicting cardiovascular risk in the general population [1]. Reverse cholesterol transport is believed to be a primary atheroprotective property of HDL particles [2]. Despite its qualities closely related to atheroprotection, the causal role of HDL in the initiation and progression of ASCVD in patients with chronic kidney disease (CKD) is completely unclear. Because of the ageing population, CKD has increased significantly in recent years. Cardiovascular disease in patients with CKD has a major impact on both human suffering and health economics. The contribution of classical atherosclerotic risk factors to the development of cardiovascular disease is less evident in patients with advanced CKD compared to patients with intact kidney function, while non-classical risk factors become more important [3].

This becomes particularly clear when one considers the lipid metabolism: Uremic dyslipidemia is characterized by hypertriglyceridemia and low HDL-C levels, whereas patients with advanced CKD rarely have elevated low-density lipoprotein cholesterol (LDL-C) levels [4]. In the healthy general population, HDL particles have strong anti-inflammatory, antioxidant and antithrombotic properties mediated by apolipoproteins, enzymes, sphingosine-1-phosphate saturated lysophosphatidylcholines and other lipids carried by these lipoproteins [5,6,7,8,9,10]. Recent studies have provided strong evidence that advanced stages of CKD with systemic oxidative stress and inflammation significantly reduce these protective activities of HDL, and dysfunctional or even pro-atherogenic forms of HDL have been identified [11,12,13,14,15,16]. 

In healthy subjects with intact renal function, the ability of HDL to promote cholesterol efflux is a predictor of cardiovascular risk, even independently of HDL-C levels [17,18]. However, despite the dramatically increased cardiovascular risk in CKD patients, neither HDL-C levels nor HDL cholesterol efflux capacity associate with prevalent ASCVD or predict future cardiovascular events in these individuals [3,19,20,21].

Important for the following discussion on the potential role of HDL particles in CKD is the notion that a major shift in the pattern of cardiovascular disease occurs in advanced CKD. Within the broad spectrum of cardiovascular disease, the relative contribution of non-atherosclerotic disease entities becomes more prominent, and the relative contribution of atherosclerotic cardiovascular disease is less important in severe CKD, when compared to individuals with intact renal function. Such non-atherosclerotic cardiovascular disease entities particularly comprise arterial calcification with subsequent left-ventricular hypertrophy, heart failure and arrhythmia [22,23]. In contrast to myocardial infarction and stroke, which are central events of atherosclerotic cardiovascular disease, these non-atherosclerotic vascular and cardiac pathologies are less consistently mediated by atherosclerotic risk factors, as described below.

## 2. CKD-Associated High Cardiovascular Risk

CKD affects more than 850,000,000 people around the globe [24]. Only a few of these people will ever develop end-stage kidney disease and subsequently require renal replacement therapy. Instead, the vast majority of CKD patients are affected by extra-renal comorbidities, among which cardiovascular disease is of particular concern. In the last two decades, numerous large epidemiological studies found an increased incidence of cardiovascular events in patients with decreased kidney function—generally reflected by low glomerular filtration rate (GFR)—as well as in patients with structural or functional abnormalities other than decreased GFR—often mirrored by albuminuria [25]. Cardiovascular risk starts rising when GFR falls below 60 mL/min/1.73 m^2^ and when albuminuria increases above 10 mg/g [25]. More advanced CKD stages are more strongly associated with future cardiovascular events than early CKD stages, and patients requiring dialysis treatment are particularly affected [26].

As a consequence, international cardiological societies—such as the European Society of Cardiology—consider patients with either moderate (GFR 30–59 m/min/1.73 m^2^) or severe (GFR < 30 mL/min/1.73 m^2^) CKD as persons at high or very high cardiovascular risk for whom intensive strategies of primary or secondary prevention are recommended [27]. In the general population, such strategies focus upon the prevention of atherosclerotic cardiovascular disease, which particularly comprise coronary heart disease, cerebrovascular artery disease and peripheral artery disease. They therefore target modifiable traditional atherosclerotic risk factors, primarily arterial hypertension, diabetes mellitus, smoking and increased LDL-C. Broad evidence confirms that targeted treatment of these traditional atherosclerotic risk factors can prevent the first or later occurrence of atherosclerotic cardiovascular disease [28].

Much less clear is the evidence that modification of the very same traditional atherosclerotic risk factor will eventually have the same benefit in patients with CKD.

### Cholesterol-Lowering Drugs in Patients with CKD

Even though data that are more comprehensive exist for LDL-C lowering in CKD patients, these data yielded an inconsistent pattern. Four large randomized clinical trials selectively recruited CKD patients in order to compare LDL-C-lowering drugs (either statin monotherapy) [29,30,31] or statin and ezetimibe as combination therapy [32] with placebo. Results from these trials suggest that the relative reduction of cardiovascular disease events via LDL-C lowering becomes smaller in patients with more advanced CKD. Most strikingly, the evidence for any effect of LDL-C lowering upon cardiovascular disease events is questionable in patients with most advanced CKD—namely, in dialysis patients. We [3,33] and others [4] have discussed a variety of reasons why prevalent CKD may affect the effects of LDL-cholesterol lowering. These comprise but are not limited to the following aspects: 

Patients with advanced CKD have relatively low baseline LDL-C levels [4]. Statins, therefore, achieve a smaller absolute LDL-C reduction than in individuals with higher baseline LDL-C, and will subsequently exert less prominent protection against atherosclerotic cardiovascular disease. Indeed, subgroup analysis from some of these randomized trials suggest that dialysis patients with higher baseline LDL-C, but not those with low baseline LDL-C, may benefit from statin treatment [31,34]. Next, in these randomized controlled trials on LDL-cholesterol lowering in CKD, patients at highest cardiovascular risk—who will particularly benefit from statin monotherapy or statin/ezetimibe combination treatment—may have been underrepresented [35]. Moreover, the primary endpoint of some of these CKD trials included a rather broad spectrum of cardiovascular events, not fully focused upon atherosclerotic cardiovascular disease. As non-atherosclerotic cardiovascular disease becomes more important in advanced CKD, statin—which mainly prevent atherosclerotic events—may become less effective. Finally, patients with CKD have an altered cholesterol metabolism. Intestinal cholesterol absorption may contribute more, and hepatic cholesterol synthesis may contribute less to plasma LDL-C, compared to individuals with intact kidney function. Therefore, in CKD patients, statin treatment—which affects cholesterol synthesis—may be less effective and ezetimibe—which affects cholesterol absorption—may be more effective for LDL-C lowering, again compared to individuals with intact kidney function. 

In conclusion, lowering LDL-C in patients with end-stage renal disease does not appear to be as effective in preventing cardiovascular events as in the general population. Importantly, lower HDL-C levels and higher triglyceride levels characterize dyslipidemia in patients with CKD. Therefore, a decrease in triglyceride levels and an increase in functional HDL particles in CKD patients could have an impact on cardiovascular risk.

## 3. What is the Physiological Function of HDL?

HDL-C is inversely associated with the risk of coronary heart disease and is a key component in predicting cardiovascular risk in the general population [1]. Reverse cholesterol transport is believed to be a primary atheroprotective property of HDL and its major protein apolipoprotein A-I (apoA-I). However, despite its activities that are linked closely to atheroprotection, the causal relationship between HDL particles and atherosclerosis remains unclear, even in the general population [1]. In a recent large study of 116,508 individuals from the general population, the association between HDL-C and all-cause mortality was U-shaped, and both extreme high- and low HDL-C concentrations associated with high mortality [36]. To date, there is no clear explanation for the “paradoxical” association of very high HDL-C and increased mortality. One hypothesis is that in individuals with extremely high HDL-C, the functional properties of HDL are altered/impaired so that HDL no longer functions normally but is more likely to cause harm. Another hypothesis is that free cholesterol transfer to HDL upon lipolysis of triglyceride-rich lipoproteins may underlie the U-shaped relationship between HDL-C and cardiovascular disease, linking HDL-C to triglyceride metabolism and atherosclerosis [37].

### 3.1. HDL Biosynthesis and Remodeling

ApoA-I is synthesized primarily in the liver and intestine. After secretion as a lipid-poor protein, apoA-I interacts with the cholesterol-phospholipid transporter ABCA1 (ATP binding cassette A1), which is expressed by hepatocytes and enterocytes to acquire lipids, thereby producing nascent HDL particles [38]. Nascent HDL particles transfer lipids to both apoB-containing particles and the plasma resident HDL pool. The unesterified cholesterol content of the nascent HDL pool is transferred to apoB-containing particles, which then redistributes to HDL for its effective esterification by lecithin-cholesterol acyltransferase (LCAT) [39]. HDL-associated cholesteryl-esters are partially transferred back to apoB-containing lipoproteins by cholesterol ester transfer protein (CETP) [40]. Although the mechanisms underlying this process are presently unknown, nascent HDL remodeling may lead to “shedding” of apoA-I from nascent lipoprotein particles as they are progressively depleted of phospholipids by phospholipid transfer protein (PLTP) to yield lipid-poor apoA-I/preß-1 HDL [41]. In turn, lipid-poor apoA-I associates rapidly with the resident HDL pool. ApoA-I synthesized by the intestine is incorporated into chylomicrons and is transferred postprandial to HDL by lipoprotein lipase-mediated hydrolysis of triglyceride-rich lipoproteins [37]. These processes are responsible for the strong inverse relationship between triglycerides and HDL-C.

Nascent HDL mobilizes excess cholesterol in an ABCA1-dependent pathway, while mature HDL removes cholesterol from lipid-loaded cells via ATP-binding cassette transporter G1 (ABCG1) [42], the scavenger receptor BI (SR-BI) [43] and passive diffusion [44]. In the reverse cholesterol transport pathway, cholesterol associated with lipoproteins is transported back to the liver either by very low-density lipoprotein (VLDL)/LDL via the LDL receptor or by HDL via SR-BI [45] and excreted into the bile through the action of ATP-binding cassette transporter G5/G8 [46]. Trans-intestinal and biliary cholesterol secretion both contribute to reverse cholesterol transport [47].

### 3.2. HDL Structure and Composition

HDL is a non-covalent quasi-spherical complex of lipids and proteins with hydrated densities from 1.063 to 1.21 g/mL. Pioneering work in 1972 by Kostner and Alaupovic identified apoA-I as the main protein component of HDL [48,49]. Since then the understanding of the composition of HDL has taken a leap forward. Several specific HDL proteins were identified and about 70% of the HDL protein content consists of apoA-I, while apoA-II makes up about 15–20% [50]. The remaining 10–15% of protein mass is composed of minor proteins, including apoCs, apoE, apoD, apoM, apoL1, apoH, apoJ and apoA-IV, enzymes (Paraoxonase 1, PAF-AH) and lipid transfer proteins such as lecithin:cholesterol acyl transferase (LCAT) and cholesteryl ester transfer protein (CETP) [51]. Compared to other serum lipoproteins, HDL is protein-rich with a protein-to-lipid ratio ranging from 1:2 in large HDL2 to 10:1 in pre-β HDL [52]. This illustrates that HDL particles are very heterogeneous and several subclasses differing in size, shape, composition and function exist.

### 3.3. HDL Subclasses

HDL exists in multiple isoforms, depending on its origin, its maturation stage and its protein and lipid composition. Therefore, HDL particles are very heterogeneous in their structure and size. Different fractions of HDL can be isolated via different methods (summarized in Table 1). The HDL isolated by ultracentrifugation can be classified according to its density into the less dense HDL2 subclass (density range of 1.063–1.125 g/mL) and the denser HDL3 subclass (density range of 1.125–1.21 g/mL) [53]. Smaller poorly lipidated HDL particles, including pre-β1 HDL, are usually not isolated via ultracentrifugation. Gradient gel electrophoresis allows for the separation of two HDL2 subclasses and three HDL3 subclasses (HDL3c, 7.2–7.8 nm diameter; HDL3b, 7.8–8.2 nm; HDL3a, 8.2–8.8 nm; HDL2a, 8.8–9.7 nm; and HDL2b, 9.7–12.0 nm) [54]. Alternatively, HDL can be separated according to surface charge and shape by agarose gel electrophoresis into α-migrating particles and pre-β-migrating particles [53]. The majority of HDL particles are α-migrating particles, whereas pre-β-migrating particles are the minority, representing small poorly lipidated pre-β1 HDL fraction as well as the large pre-β2 HDL fraction.

α- and β-migrating particles can be further separated into subclasses using 2D gel electrophoresis, which combines agarose gel with native gradient gel electrophoresis [53]. Recently, nuclear magnetic resonance (NMR) has been applied to study HDL size and particle numbers [55]. NMR allows the detection of three subclasses of HDL particles and the quantification of the particle count of HDLs. Another option to separate HDL into subclasses according to its main protein components in particles containing apoA-I and apoA-II, commonly referred to as LpA-I:A-II and particles without apoA-II, referred to as LpA-I.

## 4. CKD Profoundly Changes HDL Maturation and Metabolism

In end-stage renal disease, the level of HDL-C decreases markedly, but in contrast to the general population, no robust inverse relationship between HDL-C and cardiovascular diseases is seen [56,57]. Similar to the general population, a U-shaped association between HDL-C and all-cause and cardiovascular mortality is seen in hemodialysis patients [58]. This suggests that CKD-induced qualitative changes in HDL particles affect functionality. When compared to the general population, the LCAT-dependent conversion of preβ1-HDL to α-migrating HDL is severely delayed in CKD [13,59,60,61,62]. LCAT is activated by apoA-I and esterifies free cholesterol to cholesteryl esters to allow more efficient packaging of cholesterol for transport. The impaired LCAT-dependent conversion of lipid poor HDL in CKD might result in increased preβ1-HDL formation in patients with CKD [63]. However, a recent study did not observe a significant correlation between preβ1-HDL concentration and LCAT activity, suggesting that many factors may lead to increased preβ1-HDL levels in CKD [62]. Given that the kidney is a main site of HDL catabolism, impaired renal clearance of lipid poor HDL particles may increase preβ1-HDL in CKD. In addition, triglyceride-rich lipoproteins, which are typically elevated in CKD, promote the transfer of triglycerides to HDL particles. This process accelerates the conversion of α-migrating HDL to preβ1-HDL by CETP activity [37]. Interestingly, increased apoC-III levels in CKD are linked to clearance defects of VLDL [64,65]. ApoC-III circulates in plasma associated with apoB-containing lipoproteins and HDL, and reduces the turnover of triglyceride-rich lipoproteins mainly by inhibiting a hepatic clearance mechanism mediated by the LDL-receptor/LDL-receptor-related protein 1 axis [66].

Another factor affecting HDL-C uptake by the liver in CKD patients are plasma advanced oxidation protein products (AOPPs), which are oxidized forms of albumin or its aggregates or fragments [67,68]. AOPPs accumulate in CKD patients and are effective in blocking the major HDL receptor, SR-BI, thereby suppressing the clearance of HDL-C by the liver [68,69].

### 4.1. CKD-Associated Changes in HDL Subclasses Distribution

Studies of HDL subclasses have shown that patients with renal insufficiency have normal levels of LpA-I particles, but significantly lower levels of LpA-I:A-II particles compared to control subjects [70]. Therefore, the decrease in LpA-I:A-II particles appears to be responsible for decreased HDL-C observed in CKD patients [59]. Analysis of pre-β HDL particles in CKD patients has repeatedly shown that levels are highly increased [59,60,71]. Elevated pre-β HDL particles are present in pre-dialysis as well as in dialysis patients (Table 2). Interestingly, preβ1-HDL concentration and LCAT activity are not linked, suggesting that many factors may lead to increased preβ1-HDL levels in CKD [62]. The accumulation of pre-β HDL particles seems to be directly linked to the reduction in GFR [62]. However, the functional consequences of the pre-β particle accumulation remain to be evaluated in CKD patients. The distribution pattern of mature forms of HDL, including HDL2, HDL3 and its subclasses, have not yielded uniform results. Some groups have shown a reduction in the larger HDL2 subclass [14,72,73,74,75], others reported an increase in the HDL2 subclass [76,77,78,79]. It is clear from our analysis (Table 2) that further studies are needed to clarify the influence of CKD and dialysis on the distribution of the HDL subclasses.

### 4.2. CKD-Induced Changes in the HDL Proteome

Over the last ten years, the use of mass spectrometry has enabled the simultaneous detection of proteins in HDL isolates. This technical progress and the high sensitivity of the methods has led to studies suggesting that HDL particles could consist of more than one hundred proteins. Several research groups have performed proteomic studies on isolated HDL from patients with CKD, including work from our own group. To date, eight studies have been published, and we have summarized the most important results in Table 3. All proteomic studies investigating HDL from end-stage renal disease patients used ultracentrifugation to isolate HDL. It is important to note that different centrifugation protocols were used in these studies, including single stage, two-stage density gradient or sequential ultracentrifugation methods. The number of identified proteins ranged from 35 to 326, indicating a very high degree of heterogeneity in these studies (Table 3). Since the quality controls for isolated HDL are not standardized and important specifics of HDL such as size distribution, purity, apoA-I and cholesterol content have not been defined, it is difficult to compare the individual results. The heterogeneities are probably due to differences in the isolation methodology and further purification of the HDL isolates as well as different mass spectrometry technology. The first proteomic study investigating HDL isolated from CKD patients was published in 2011 [11]. HDL was isolated from 27 end-stage renal disease patients on maintenance hemodialysis and used for proteomic assessment in comparison to HDL isolated from healthy controls. The results suggested a significant reduction in apoA-I, apoA-II, apoC-I and apoM and an increase in apoC-III, apoA-IV, α-1-antitrypsin, retinol-binding protein 4 and α-2 catenin. Other studies have confirmed these results and provided evidence that the content of apoA-I and apoA-II of HDL was decreased [13,80], and the content of apoC-III and serum amyloid a (SAA) in HDL from CKD patients is increased [81,82,83]. It was found that the HDL cholesterol efflux capacity of uremic HDL is significantly reduced. The overall conclusion of these studies is that changes in the composition of HDL in CKD is associated with significant functional impairment of HDL.

Since then, other studies have added to the list of potentially HDL-associated proteins. In 2012, Weichhart et al. performed a HDL proteomic study on a small cohort of end-stage renal disease patients on maintenance hemodialysis [12]. HDL from end-stage renal disease patients was enriched in apoC-II, SAA, surfactant protein-B and protein AMBP (Table 3). The authors suggested, that SAA is involved in the loss of anti-inflammatory activity of HDL. A specific relationship between the accumulation of surfactant protein-B and the protein AMBP with observed functional impairment of HDL remains to be established.

A very similar approach was followed by Mange et al. [84] and Shao et al. [85], who also compared a small cohort of end-stage renal disease patients with healthy controls. They identified 122 and 63 proteins, respectively, in isolated HDL and identified several proteins to be enriched or decreased. Most notably, they found that apoC-III, apoA-IV and SAA were enriched, while the report from Shao et al. additionally identified apoA-I, apoA-II, apoM and paraoxonase 1 to be decreased. The authors concluded that comprehensive remodeling of HDL occurs in uremic subjects, but unfortunately the functional properties of HDL were not investigated.

Kopecky et al. investigated the impact of kidney transplantation on the proteomic composition of HDL in a small cohort consisting of 28 transplant patients [82]. Their proteomic assessment identified 80 proteins in isolated HDL, several of them claimed to be specific enriched in HDL from transplant patients (Table 3). Surprisingly, despite the present low-grade inflammation (judged by serum C reactive protein values) an increase in SAA has not been detected by proteomic assessment. However, by using ELISA the study indeed confirmed an increased content of SAA on HDL particles when compared to HDL isolated from healthy controls.

Rubinow et al. investigated a large CKD cohort of 509 pre-dialysis patients with a broad range of estimated glomerular filtration rates, corresponding to CKD stages I-V [83]. Proteomics identified 38 proteins within isolated HDL and their data analysis focused on proteins which increased or decreased with declining glomerular filtration rates (GFR). Interestingly, they found that for each 15-mL/min per 1.73 m2 lower GFR, the content of retinol binding protein 4 and apoC-III was increased, while the contents of apoL1, CETP and vitronectin were decreased within HDL isolates. Overall, they concluded that a moderate alteration in the HDL proteome occurs as another facet of the metabolic derangements attendant to GFR loss. These results indicate that major changes in the HDL proteome already occur at pre-dialysis. This aspect was investigated by Wang et al., who focused on changes in the HDL proteome when initiating hemodialysis [86]. The study cohort consisted of 110 participants with advanced CKD and 143 participants that initiated hemodialysis within the last year. They quantified 38 proteins within their HDL isolates and found eight proteins with a greater relative abundance after hemodialysis was initiated. The identified HDL constituents were mainly markers of inflammatory, atherosclerotic, and lipid metabolism pathways, namely SAA1, SAA2, hemoglobin-b, haptoglobin-related protein, CETP, PLTP and apoE. None of these HDL-associated proteins, with the exception of CETP, were associated with lower GFR in a recent study of non-dialysis patients [83]. These studies combined suggest that CKD and hemodialysis might uniquely affect the HDL proteome, thereby generating different versions of dysfunctional HDL. The most recent study examining the HDL proteome in patients with end-stage renal failure was published in 2019. The study analyzed a very small cohort of nine end stage renal disease patients on hemodialysis and eight controls [87]. In HDL isolates, 326 proteins were identified—ten proteins were found to be upregulated and nine to be downregulated (Table 3). Surprisingly, their analysis did not match any of the alterations described in other reports, such as an increase in SAA or apoC-III or a decrease in apoA-I, apoA-II as described above. Overall, the most common changes in the composition of HDL in CKD observed in most studies are the increase in SAA1 and apoC-III, while the activity of HDL-associated paraoxonase 1 is significantly impaired.

SAA1 is a major acute-phase protein, secreted predominantly by the liver during the acute phase of inflammation. SAA1 is primarily bound to HDL in the circulation. Research has consistently shown that serum levels of SAA1 and HDL-associated SAA1 are increased in patients with CKD [88]. SAA has been regarded as a pro-inflammatory and atherogenic mediator [89,90,91,92]. Previous studies reported that HDL artificially enriched in SAA was less potent to inhibit the oxidation of low-density lipoprotein [89], and induced the production of the chemokine MCP-1 in human monocytes [81,93], leading to migration and tissue infiltration of monocytes into atherosclerotic plaques [91]. Furthermore, SAA-enriched HDL was shown to bind to biglycans thereby reducing the athero-protective function of HDL [94]. Recently, however, serious doubts have been raised as to whether SAA is truly pro-inflammatory. The majority of published reports on the pro-inflammatory activities of SAA used recombinant human SAA expressed in *Escherichia coli*. Two studies convincingly demonstrated that pro-inflammatory activities of recombinant SAA are not shared by the endogenous protein in the circulation [95,96]. Interestingly, recent reports have even pointed towards a beneficial role of SAA. Cheng et al. showed that SAA promotes LPS clearance and suppressed LPS-induced inflammation and tissue injury [97]. Furthermore, SAA produced by the intestine in response to microbiota serves as a systemic signal to neutrophils to restrict aberrant activation and decreasing inflammatory tone [98]. An interesting recent study has proposed another role for SAA. The authors hypothesized that high SAA levels are necessary to reroute HDL for rapid removal of membranes from dead cells as a first line of defense at injured sites [99]. Overall, the picture is ambivalent as to whether SAA perform anti-inflammatory or pro-inflammatory functions. Therefore, further studies are needed to clarify the true role of SAA as an important component of HDL particles in inflammation.

ApoC-III is an 8.8-kDa highly glycosylated protein mainly produced in the liver and to a lesser extent in the intestine. ApoC-III is a physiological inhibitor of lipoprotein lipase and is highly associated with hypertriglyceridemia and a strong independent predictor of CVD risk [100,101]. Increased apoC-III plasma concentrations in CKD patients have been shown to be a consequence of the disturbed catabolism of apoC-III [65]. Studies have shown that the content of HDL-associated apoC-III increases with decreasing kidney function and only the content of HDL without apoC-III was associated with a lower cardiovascular disease risk [102]. These observations support the concept of “dysfunctional HDL” that might be caused by the excessive incorporation of apoC-III. ApoC-III enrichment in HDL impairs HDL-mediated cholesterol efflux capacity [11,103] and can strongly influence immune cell response by promoting inflammation and organ damage through alternative inflammasome activation [104]. Sterile inflammation by activation of the inflammasome is a key step in the pathogenesis of a variety of diseases, such as cardiovascular disease and CKD. ApoC-III induces NLRP3-inflammasome-driven inflammation in human monocytes via caspase 8 and the dimerization of toll-like receptors 2 and 4. Human monocytes activated by apoC-III critically impair endothelial regeneration and promote renal damage in animal models [104]. ApoC-III has been demonstrated to have direct atherogenic properties by stimulating the adhesion of blood monocytes to endothelial cells and inducing the production of inflammatory mediators, including interleukin-1β [105,106]. Furthermore, apoC-III promotes smooth muscle cell proliferation via Akt signaling pathway mediated by reactive oxygen species in vitro, leading to aggravated restenosis and atherogenesis [107]. It should be noted that recombinant apoC-III appears to act in an anti-allergic manner. For example, apoC-III reduces the activation of eosinophils, which play a key role in allergies [108,109].

Paraoxonase 1 is a circulating Ca2+-dependent esterase/lactonase with a molecular mass of about 43 kDa [110]. Paraoxonase 1 is mainly synthesized by the liver and associates with HDL. The activity of paraoxonase 1 is sensitive to alterations in the HDL particle [111,112]. The natural substrates for PON1 are lactones and lipophilic derivatives derived from it. Studies have repeatedly shown that paraoxonase 1 enzymatic activity as well as its content is reduced in CKD [13,113,114,115,116] (Table 3). As part of HDL, paraoxonase 1 has been suggested as a major factor improving HDL functionality, especially the anti-inflammatory activity of HDL. Recombinant paraoxonase 1 inhibits MCP-1 expression in cultured endothelial cells [117]. Transgenic mice expressing human paraoxonase 1 developed less atherosclerosis on a high-fat and high-cholesterol diet compared to wild type mice [118]. Paraoxonase 1 may increase macrophage cholesterol efflux by increasing the lysophosphatidylcholine content, which stimulates the activity of ABCA1, HDL binding to the cells, and the cholesterol uptake via HDL [119]. During CKD-associated inflammation, apoA-I is displaced from HDL by SAA, resulting in decreased paraoxonase 1 activity.

## 5. HDL-Cholesterol Efflux Capacity, a Key Functional Metric of HDL

A key component in the development of atherosclerosis is the overloading of macrophages with cholesterol in the arterial wall, resulting in foam cell formation and activation of macrophages. There is robust evidence of an inverse association between plasma HDL-cholesterol concentrations and the risk of cardiovascular disease in the general population, leading to the assumption that HDL protects from cardiovascular disease [7]. HDL has been traditionally regarded as the key component in reverse cholesterol transport, by virtue of its capacity to remove excess cholesterol from lipid-laden macrophages, peripheral tissue and from circulating immune cells and endothelial cells [2,120,121,122,123]. Different in vitro tests have been developed to measure the ability of HDL to promote cholesterol efflux from macrophages. In the best established assay (Figure 1), the mouse macrophage cell line J774 is enriched with radioactively or fluorescently labeled cholesterol and stimulated with cAMP (or loaded with acetylated LDL) to induce expression of cholesterol transporters. In addition, the mouse macrophage cell line RAW or the human cell line THP-1 are also used for cholesterol efflux tests. Subsequently, isolated HDL or apoB-depleted serum from patients is added to the cell medium to assess HDL cholesterol efflux capacity. ApoB-depleted serum obtained by precipitation is often preferred because this fast and simple method removes LDL and VLDL from the serum in a much gentler way when compared to ultracentrifugation, which removes some of the many functionally important proteins from the HDL particle. In addition, poorly lipidated HDL subfractions are lost when HDL is isolated by density ultracentrifugation. ApoB-depleted serum abolishes the transfer of labeled cholesterol from cells to these apoB-containing lipoproteins, and exclusively analyzes the cholesterol transport capacity to HDL particles. Moreover, apoB-depleted serum can be stored for years in the absence of cryoprotectants, which is not possible with isolated HDL [124]. After incubation, the efflux of labeled cholesterol from cells to the medium is quantified and reflects the cholesterol transport mediated by ATP-binding cassette transporter A1 (ABCA1), ABCG1, and scavenger receptor class B, member 1 (SR-BI) and aqueous diffusion (Figure 1). To ensure a reliable measurement of the cholesterol efflux capacity, standardized criteria must be applied, since the test principle can be modified in many ways.

### 5.1. HDL Cholesterol Efflux Capacity Is a Robust Predictor of Cardiovascular Events in the General Population

A causal role of HDL in the development of atherosclerosis is still under debate. A more important factor may be represented by the HDL cholesterol efflux capacity, the first and most likely rate-limiting step of the reverse cholesterol transport. HDL cholesterol efflux capacity assessed in apoB-depleted serum, is inversely associated with early, asymptomatic atherosclerotic vascular disease in the general population [126] and with incident cardiovascular events among the general population [17,18,127,128]. Of particular note, HDL-cholesterol efflux capacity remains strongly associated with cardiovascular disease after adjustment for confounders, even after adjusting for HDL-C levels.

### 5.2. HDL-Cholesterol Efflux Capacity in CKD Patients

HDL isolated from CKD patients depicts a reduced cellular cholesterol efflux capability [11,13,14,15], which is linked to a depletion of HDL-associated apoA-I, apoA-II, and phospholipids, and increased apoC-III and SAA (Figure 2), all factors that are known to modulate the cholesterol acceptor capability of HDL [103,129,130]. Despite these pronounced compositional changes in uremic HDL and subsequent decreased cholesterol efflux capacity, this did not necessarily lead to clear associations with cardiovascular outcomes. For example, the HDL cholesterol efflux capacity of apoB-depleted serum was not associated with cardiovascular outcome in the CARE FOR HOMe study, a prospective cohort study with pre-dialysis CKD patients [19,116] and in the German Diabetes Dialysis Study (4D Study) [21]. In the Dallas Heart Study , a population-based cohort of participants without widespread cardiovascular disease, a higher cholesterol efflux capacity was paradoxically even associated with an increase in cardiovascular mortality among participants with a baseline eGFR < 60 mL/min/1.73 m^2^ [20]. Interestingly, one study among renal allograft recipients reported that cholesterol efflux capacity strongly predicted kidney graft failure independent of plasma HDL-C levels [131]. Therefore, increasing HDL function might be a treatment target for the prevention of graft failure. An example of a strategy to improve HDL function/anti-inflammatory properties was recently reported, showing that IL-1 blockade improves the anti-inflammatory and antioxidative properties of the HDL-containing fraction of plasma in patients with stages 3–5 CKD, including those on maintenance hemodialysis [132].

## 6. CKD-Associated Changes of Other HDL Functions

In addition to the cholesterol efflux capacity, HDL shows additional properties that are considered antiatherogenic. As depicted in Figure 2, the most prominent CKD-induced changes in HDL composition, are phospholipid depletion, reduced apoA-I, apoA-II and paraoxonase 1 levels and enrichment with proinflammatory proteins SAA and apoC-III. These alterations as well as post-translational and oxidative modifications affect HDL functions such as cholesterol efflux capacity [11,13,14,15,16], anti-inflammatory capacity [12,15] as well as endothelial protective activities [15,133]. The major protein of HDL, apoA-I as well as paraoxonase 1, that cotransports with HDL in plasma are well known to have antioxidant properties [134]. Consequently, HDL has the ability to inhibit LDL oxidation thereby reducing the atherogenicity of these lipoproteins. CKD profoundly impairs HDL-associated paraoxonase activity and antioxidative capacity of HDL [13,61,88,114,135]. A number of reactive uremic toxins accumulate in the plasma of CKD patients [136] and structurally alter the HDL particles and influence HDL function. Cyanate, an electrophilic reactive species in equilibrium with urea, modifies proteins post-translationally by a process called carbamylation [137]. In uremia, elevated concentrations of carbamylated HDL are detected, which strongly alter the structure and function of the HDL particles [138,139,140,141]. For example, carbamylation reduces HDL-associated paraoxonase activity and the ability of HDL to activate LCAT [141]. In addition, carbamylated HDL can increase foam cell formation via a HDL receptor (SR-BI)-mediated pathway [140].

However, these other compositional and functional properties of HDL do not seem to consistently predict cardiovascular outcome in CKD patients, similar to cholesterol efflux capacity. Lower HDL-paraoxonase activity and higher HDL-SAA content are predictors of a negative outcome in pre-dialysis patients of the CARE FOR HOMe study in univariate Cox regression analyses [116]. However, after adjustment for traditional cardiovascular and renal risk factors and systemic inflammatory markers, none of these parameters significantly associates with a negative outcome [116]. On the other hand, paraoxonase activity showed an association with 12-month mortality in patients on maintenance hemodialysis [114]. Therefore, further studies are needed to draw firm conclusions.

## 7. Conclusions

Significant abnormalities in the composition of HDL are observed, which all lead to the formation of dysfunctional HDL. Although significant abnormalities in the HDL-induced cholesterol efflux capacity and other functionalities are seen in patients with CKD, this does not necessarily lead to clear associations with cardiovascular outcomes. However, one study among renal allograft recipients reported that cholesterol efflux capacity strongly predicted kidney graft failure independent of plasma HDL-C levels. Therefore, increasing HDL function might be a treatment target for the prevention of graft failure. Moreover, the fact remains that the overexpression of apoA-I, the major protein component of HDL particles in preclinical models, has a variety of positive effects on inflammation and even promotes the regression of atherosclerosis and diabetes. Novel HDL “functionality enhancing” therapies could reduce the decline in renal function. It remains to be seen whether these concepts can be translated into new therapeutic interventions for CKD patients.

## Figures and Tables

**Figure 1 biomolecules-10-01348-f001:**
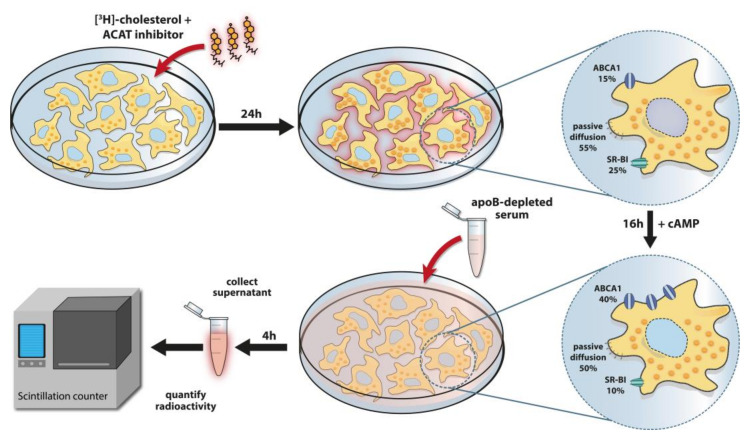
Principle of the cholesterol efflux assay. J774 macrophages are cultivated in multiwell plates to form a monolayer. The cells are then treated for 24 h with an ACAT (acyl coenzyme A: cholesterol acyltransferase) inhibitor and radiolabeled cholesterol ([^3^H]-cholesterol). The ACAT inhibitor prevents cholesterol esterification and the added cholesterol remains cell-associated as free cholesterol. On the following day, the cells are treated with cyclic adenosine monophosphate (cAMP) for 16 h to stimulate the expression of the cholesterol exporter ABCA1. The cholesterol efflux in unstimulated macrophages is mediated to 15% by ABCA1, 25% by SR-BI and 55% by passive diffusion (includes ABCG1-mediated efflux). By cAMP treatment, the ABCA1-dependent cholesterol efflux triples to about 40%, while passive diffusion accounts for 50% and SR-BI-mediated efflux for 10% [125]. Human serum shows a depletion of lipoproteins containing apoB100 (mainly VLDL, LDL) using polyethylene glycol. After extensive rinsing of the cells, apoB-depleted serum (containing all HDL subclasses) is added to the [^3^H]-cholesterol-labeled macrophages at a concentration of 2.8%. After 4 h, the [^3^H]-cholesterol that has passed from the cells into the supernatant is quantified by liquid scintillation counting.

**Figure 2 biomolecules-10-01348-f002:**
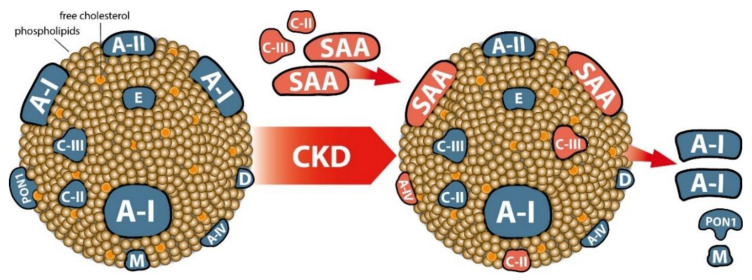
Most frequently identified changes in the proteome of HDL in CKD patients. Approximately 70% of the HDL protein mass is comprised of apoA-I (A-I), while apoA-II(A-II) comprises about 15–20% [50]. The remaining 10–15% of protein mass is composed of less abundant proteins, including apoC-III, apoC-II, apoE, apoD, apoM, apoA-IV, as well as enzymes such as paraoxonase 1 (PON1) and lipid transfer proteins, including lecithin:cholesterol acyl transferase and cholesteryl ester transfer protein [50]. To simplify the illustration only the major constituents of HDL are shown. In CKD, a specific remodeling of the HDL particle occurs depending on the stage of CKD and the vintage of dialysis treatment. The most noticeable change in the composition of HDL in CKD is the accumulation of serum amyloid a (SAA), especially SAA1, together with the enrichment in apoC-II and apoC-III. The accumulation of these proteins is accompanied by a loss of apoA-I, apoA-II, apoM and a decrease in the mass and enzymatic activity of paraoxonase 1 (PON1).

**Table 1 biomolecules-10-01348-t001:** High-density lipoprotein (HDL) subclasses depending on the isolation method.

Density (Ultracentrifugation)	δ g/mL
HDL2	1.063–1.125
HDL3	1.125–1.210
**Size (Electrophoresis)**	**nm**
HDL2b	9.7–12.0
HDL2a	8.8–9.7
HDL3a	8.2–8.8
HDL3b	7.8–8.2
HDL3c	7.2–7.8
**Charge and Size (2D Electrophoresis)**	**particles**
Preβ-HDL	preβ1, preβ2
α-HDL	α1, α2, α3, α4
Preα-HDL	preα1, preα2, preα3
**Composition (Antibody-Based)**	
LpA-I	apoA-I
LpA-I:A-II	apoA-I + apoA-II
**Size + Particle Number (Nuclear Magnetic Resonance (NMR))**	**nm**
Large HDL	8.8–13.0
Medium HDL	8.2–8.8
Small HDL	7.3–8.2

**Table 2 biomolecules-10-01348-t002:** Summary of studies investigating HDL subpopulations in chronic kidney disease.

Study	Cohort	Method	Control	Chronic Kidney Disease (CKD)	Hemodialysis (HD)
**Samuelsson et al. 2002** [70]	CKD, *n* = 45 Controls, *n* = 45	Immuno-absorption	mg/dL LpA-I = 34.7 ± 7.1 LpA-I:A-II=103.4 ± 18.0	mg/dL LpA-I = 32.6 ± 5.3 LpA-I:A-II = 93.6 ± 14.5	-
**Calabresi et al. 2015** [59]	CKD, *n* = 50 HD, *n* = 198 Controls, *n* = 40	Immuno-absorption + Native gel electrophoresis	mg/dL LpA-I = 50.1 ± 13.2 LpA-I:A-II = 84.6 ± 12.7 % of total HDL protein preβ-HDL = 13.1 ± 3.2 size (nm) HDL2 = 11.2 ± 0.3 HDL3 = 8.8 ± 0.3	mg/dL LpA-I = 55.2 ± 15.9 LpA-I:A-II = 64.0 ± 14.4 % of total HDL protein preβ-HDL = 15.8 ± 4.7 size (nm) HDL2 = 11.1 ± 0.2 HDL3 = 8.9 ± 0.2	mg/dL LpA-I = 43.1 ± 12.8 LpA-I:A-II = 49.2 ± 13.6 % of total HDL protein preβ-HDL = 17.1 ± 4.7 size (nm) HDL2 = 11.2 ± 0.4 HDL3 = 8.8 ± 0.4
**Holzer et al. 2015** [13]	CKD, *n* = 24 Controls, *n* = 20	Ultracentrifugation + Native gel electrophoresis	% of total protein HDL2 = 44.8 ± 5.7 HDL3 = 55.2 ± 5.8	-	% of total protein HDL2 = 40.7 ± 6.1 HDL3 = 59.4 ± 6.0
**Homma et al. 2013** [76]	CKD, *n* = 40 Controls, *n* = 40	sequential ultracentrifugation	mg/dL cholesterol HDL2 = 21.8 ± 6.9 HDL3 = 21.5 ± 4.8	-	mg/dL cholesterol HDL2 = 30.6 ± 12.3 HDL3 = 17.6 ± 4.5
**Kuchta et al. 2019** [62]	3 CKD groups Stage 3a, *n* = 17 Stage 3b, *n* = 34 Stage 4, *n* = 17	ELISA	-	pre-β1 HDL (mg/dL) Stage 3a, 1.85 ± 0.20 Stage 3b, 2.20 ± 0.21 Stage 4, 2.70 ± 0.22	-
**Gille et al. 2019** [71]	CKD, *n* = 16 Controls, *n* = 16	ELISA	mg/mL preβ1-HDL = 16 ± 3	mg/mL preβ1-HDL = 23 ± 1	-
**Miida et al. 2003** [60]	CKD, *n* = 45 Controls, *n* = 45	2D Native gel electrophoresis	% of apoA-I preβ1-HDL = 5.0 ± 2.0 preβ2-HDL = 4.6 ± 2.5 preβ3-HDL = 0.9 ± 0.5 HDL2b = 20.0 ± 13.5 HDL2a = 37.4 ± 10.0 HDL3 = 27.0 ± 9.7	-	% of apoA-I preβ1-HDL = 13.5 ± 3.5 preβ2-HDL = 6.0 ± 2.1 preβ3-HDL = 1.0 ± 0.5 HDL2b = 22.5 ± 5.5 HDL2a = 35.5 ± 5.8 HDL3 = 21.5 ± 4.1
**Alabakovska et al. 2002** [79]	CKD, *n* = 42 HD, *n* = 63 Controls, *n* = 345	Native gel electrophoresis	percent distribution HDL2b = 50.0% HDL2a = 45.5% HDL3a = 4.5% HDL3b = 0.0% HDL3c = 0.0%	percent distribution HDL2b = 16.5% HDL2a = 62.0% HDL3a = 21.5% HDL3b = 0.0% HDL3c = 0.0%	percent distribution HDL2b = 30.0% HDL2a = 67.0% HDL3a = 3.0% HDL3b = 0.0% HDL3c = 0.0%
**Stefanovic et al. 2017** [78]	CKD, *n* = 19 PT, *n* = 19	Native gel electrophoresis	% of total HDL protein HDL2b = 48.6 ± 4.9 HDL2a = 22.6 ± 2.3 HDL3a = 14.0 ± 2.2 HDL3b = 7.5 ± 1.5 HDL3c = 7.2 ± 1.5	% of total HDL protein HDL2b = 39.3 ± 5.4 HDL2a = 21.3 ± 2.1 HDL3a = 16.8 ± 2.6 HDL3b = 10.1 ± 2.1 HDL3c = 12.6 ± 6.5	-
**Miljkovic et al. 2018** [75]	CKD, *n* = 21 HD, *n* = 56 Controls, *n* = 20	Native gel electrophoresis	% of total HDL protein HDL2b = 54.0 ± 9.7 HDL 2a = 18(16.1–21.5) HDL3a = 11.0 ± 3.2 HDL3b = 6.2(3.8–8.4) HDL3c = 10.4 (6.0–11.4)	% of total HDL protein HDL2b = 46.0 ± 9.8 HDL 2a = 20 (17.9-23.6) HDL3a = 13.0 ± 2.8 HDL3b = 7.4 (5.4–9.3) HDL3c = 13.9 (5.2–16.9)	% of total HDL protein HDL2b = 44.0 ± 11.6 HDL 2a = 22 (18.1-24.1) HDL3a = 14.0 ± 4.5 HDL3b = 7.7 (6.6–10.9) HDL3c = 10.2 (6.8–14.8)
**Soto-Miranda et al. 2012** [77]	CKD, *n* = 40 Controls, *n* = 40	Ultracentrifugation + Native gel electrophoresis	% of total HDL protein HDL2b = 15.7 ± 5.7 HDL2a = 8.8 ± 1.9 HDL3a = 24.1 ± 2.7 HDL3b = 20.6 ± 2.9 HDL3c = 31.9 ± 7.3	% of total HDL protein HDL2b = 23.5 ± 5.9 HDL2a = 11.6 ± 1.9 HDL3a = 24.6 ± 2.4 HDL3b = 15.6 ± 2.5 HDL3c = 24.8 ± 5.7	-

**Table 3 biomolecules-10-01348-t003:** Summary of studies investigating the HDL proteome in chronic kidney disease.

Study	Cohort	Isolation Method	Detected Proteins	Proteins Upregulated	Proteins Downregulated	Validation Test	Functional Assessment
**Holzer et al.** **2011** [12]	Control, *n* = 19 HD, *n* = 27	density gradient ultracentrifugation	35	apoC-III, SAA1, SAA4, apoC-II, apoA-IV, A1At, RBP4, TTR, a2CAT	apoA-I, apoA-II, apoC-I, apoM	Results for albumin, Lp-PLA2, A1AT, ApoAIV, ApoA-I, RBP4, TTR and SAA1 confirmed by immunoblot.	Total cholesterol efflux ↓ scavenger receptor BI (SR-BI)-specific cholesterol efflux ↓ ATP-binding cassette transporter A1 (ABCA1)-specific cholesterol efflux ↔ Macrophage net cholesterol efflux ↓ HDL-associated Lp-PLA2 activity ↑
**Weichhart et al.** **2012** [13]	Control, *n* = 10 HD, *n* = 10	sequential ultracentrifugation	49	apoC-II, SAA, SP-B, AMBP	-	Replica cohort of 12 control and 14 HD used to confirm MS result by immunoblot for TF, Sp-B, PEDF, SAA, apoC-II, apoA-I	HDL anti-inflammatory activity ↓ HDL anti-oxidative activity ↓
**Mange et al.** **2012** [84]	Control, *n* = 7 HD, *n* = 7	sequential ultracentrifugation	122	apoA2, apoC3, AMBP, apoD, apoC2, B2MG, SAA4, apo(a), RBP4, ApoC1, LCAT, ApoA4, ApoE, SAA, ApoM, PON1, ApoC4, ApoL1, ApoB100	ST, C3, FIB, HG, Igα, A2MG, CFH, Igμ, FIBR, HP, KIN1, PT, HRG, ITIH4, VTN, AT3, CLUS, Igλ	Results for apoC2, apoC3, ST, HG confirmed in validation cohort.	not performed
**Shao et al.** **2015** [85]	Control, *n* = 20 HD, *n* = 40	sequential ultracentrifugation	63	AMBP, B2MG, CFD, CST3, PTGDS, RBP4, SAA1, CST3, AMBP, CFD, PTGDS, SAA4, TTR, ApoCII, apoCIII, A1GP2, apoAIV,Igk, SP-B, Igλ, SP-B	apoA-I, apoA-II, apoL-I, apoM, PON1, VTN.	Shotgun proteomics used for identification of proteins, followed by SRM to quantify and validate.	not performed
**Kopecky et al.** **2015** [82]	Controls, *n* = 15 HD, *n* = 14 KTxpoor, *n* = 14 KTxgood, *n* = 14	density gradient ultracentrifugation	80	Ktxpoor and HD: AMBP, B2MG, RBP4, Igγ3, FIBR, CFD, ZA2GP. Ktxpoor: B2GP1, LRA2GP, apo(a), CAMP, A1CT, ANG, PC1, CYS, SHDP, VDBP, A1AGP	-	Enrichment of SAA and SP-B in Ktxgood, Ktxpoor and HD quantified with ELISA.	Cholesterol efflux ↓ vs. Ktxgood Arylesterase activity ↓ vs. Ktxgood, Ktxpoor, HD; Leukocyte cholesterol content ↑ vs. Ktxgood, Ktxpoor, HD
**Rubinow et al.** **2017** [83]	CKD, *n* = 538. 5 groups: eGFR >60, *n* = 92 eGFR = 45-60, *n* = 91 eGFR = 30-45, *n* = 106 eGFR = 15-30, *n* = 102 eGFR < 15, *n* = 34	2-step density gradient ultracentrifugation	38	RBP4, apoC3↑	ApoL1, CETP, VN↓	-	not performed
**Wang et al.** **2018** [86]	Pre-dialysis, *n* = 110 Hemodialysis, *n* = 143	2-step density gradient ultracentrifugation	38	SAA2, HBB, SAA1, HPR, CETP, PLTP, ApoE	-	-	Cholesterol efflux ↓ vs. pre-dialysis
**Florens et al.** **2019** [87]	Control, *n* = 8 HD, *n* = 9	sequential ultracentrifugation	326	UDP 1, B2MG, SP-B, AMBP, IGF2, IGHA2, IGLC2, HLA-B, CFD, ITIH4	GUCA, CAPN1, KRT16, RAB6B, GM2A, PTGDS, SCGB, PRDX3, SCF2	-	not performed

Ktxgood: kidney transplant patient with good graft function; Ktxpoor: kidney transplant patient with poor graft function. A1AT, α-1-antytrypsin; A1AGP, α-1-acid-glycoprotein 2; A1CT, α-1-antichymotrypsin; A1GP2, α-1-glycoprotein 2; A2MG, α-2-macrolobulin; AMBP, Protein AMBP; ANG, angiotensinogen; Apo, apolipoprotein; AT3, antithrombin-III; B2GP1, β-2-glycoprotein 1; B2MG, β-2-microglobulin; C3, complement C3; CAMP, cahelicidin antimicrobial peptide; CAPN1, Calpain-1 catalytic subunit; CETP, cholesterylester transfer protein; CFD, complement factor D; CLUS, clusterin; CYC, cystatin C; FIBR, fibinogen alpha chain; GM2A, Ganglioside GM2 activator; GUCA, Guanylin; HBB, hemoglobin-β; HG, haptoglobin; HRG, histidine-rich glycoprotein; HLA-B, HLA class I histocompatibility antigen, B-58 alpha chain; HPR, haptoglobin related protein; IGF2, Insulin-like growth factor II; ITIH4, inter-alpha-trypsin inhibitor; IGHA2, Immunoglobulin heavy constant alpha 2; IGLC2, Immunoglobulin lambda constant 2; Igα, immungobulin alpha-1 chain C; Igμ, immungobulin mu chain C; Igγ, Immunglobulin gamma; Igλ, Immunglobulin lamba; KIN1, kininogen-1; KRT16, Keratin, type I cytoskeletal 16; LRA2GP, leucine-rich-α-2-glycoprotein; LCAT, Phosphatidylcholine-sterol acyltransferase; PC1, plasma protease C1 inhibitor; PEDF, pigment epithelial derived factor; PLTP, phospholipid transfer protein; PON1, paraoxonase 1; PT, prothrombin; PTGDS, prostaglandin-H2-D isomerase; PRDX3, Thioredoxin-dependent peroxide reductase, mitochondrial; RAB6B, Ras-related protein Rab-6B; RBP4, retinol-binding protein 4; SAA, serum amyloid A; SCF2, Solute carrier family 2, facilitate; SCGB, Secretoglobin family 3A member 2; SHDP, SH3 DB glutamic acid-rich-like protein 3; SP-B, surfactant protein B; ST, serotransferrin; TF, transferrin; TTR, transthyretin; UDP1, UDP-glucose: glycoprotein glucosyltransferase 1; VDBP, vitamin D binding protein; VTN, vitronectin; ZA2GP, zinc-α-2-glycoprotein.

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
