# Peer review of "Current Understanding of the Relationship of HDL Composition, Structure and Function to Their Cardioprotective Properties in Chronic Kidney Disease"

_biomolecules, 2020, doi:10.3390/biom10091348_

Round 1

Reviewer 1 Report

The aim of the review was to summarize the recent studies on the relationship between  HDL composition, structure and function and cardioprotective properties in CKD patients.

It is a well written and organized review, I have no comments and I suggest to publish  it. 

Author Response

Thank you for your kind comments and for recommending our manuscript for publication.

Reviewer 2 Report

The paper by Marsche and coworkers reviews the current knowledge of the relationship between the cardioprotective properties of HDL and cardiovascular risk in chronic kidney disease. This is a relevant topic since HDL function could be a future target for prevention of cardiovascular events in patients with severe kidney disease. The authors have a long history in the study of the function of lipoproteins in the setting of kidney disease and demonstrate a profound knowledge of the topic. The review is well written and addresses the main aspects of the relationship between HDL function, cardiovascular risk and kidney disease. I only have some minor comments.

  • The main concern is an error in the assignment of the reference numbers from reference 17. From this point on, all the numbers are displaced and correspond to the previous reference in the bibliography section. The cause of this error is probably due to the redundancy of two sentences that are very similar and include almost the same references. The first sentence is on lines 48-51 (references 12-16), and the second sentence, which repeats the same idea, is on lines 51-54 (references 12-17). I suggest deleting one of the two sentences.
  • Line 147, please, correct “unesterfied”.
  • Line 211, I think that “HDL metabolism” should be changed to “HDL catabolism”.
  • Line 223. I suggest to change in this title “HDL structure” to “HDL subclasses distribution”. This paragraph does not really deal with the structure of HDL particles or their apolipoproteins but with the distribution or classification of HDL subfractions.
  • The numbering of Tables is wrong in pages 8, 10 and 11 (lines 266, 280, 306 and 358).
  • I think it should be good to specify that apoC-III in the physiological inhibitor of LpL (line 333-335).
  • In my opinion, Figure 1 does not match well with the legend. The incubation with the ACAT inhibitor is not shown in the figure. And the relative distribution of efflux shown in the draws shown at the right side seems to indicate that cAMP does not induce ABCA1 from 15% to 40%; instead, it seems that is apoB-depleted serum that increases such expression (which is incorrect). This part of the figure is confusing and should be improved.
  • Line 446. As far as I know, the antioxidant role of apoA-II has not been demonstrated, I suggest delete apoA-II from this sentence.

Author Response

Thank you for your interest in our review article and the very helpful comments.

We thank the reviewer for drawing our attention to the incorrect assignment of the reference numbers. This has been corrected accordingly.

Line 147, please, correct “unesterfied”.

This has been corrected accordingly (Line 143).

Line 211, I think that “HDL metabolism” should be changed to “HDL catabolism”.

This has been corrected accordingly (Line 209).

Line 223. I suggest to change in this title “HDL structure” to “HDL subclasses distribution”. This paragraph does not really deal with the structure of HDL particles or their apolipoproteins but with the distribution or classification of HDL subfractions.

We agree. This has been corrected accordingly (Line 221). 

The numbering of Tables is wrong in pages 8, 10 and 11 (lines 266, 280, 306 and 358).

Thank you for this comment. This has been corrected accordingly.

I think it should be good to specify that apoC-III in the physiological inhibitor of LpL (line 333-335).

According to the reviewer's suggestion we have now specified that apoC-III is a physiological inhibitor of LPL (Line 338).

In my opinion, Figure 1 does not match well with the legend. The incubation with the ACAT inhibitor is not shown in the figure. And the relative distribution of efflux shown in the draws shown at the right side seems to indicate that cAMP does not induce ABCA1 from 15% to 40%; instead, it seems that is apoB-depleted serum that increases such expression (which is incorrect). This part of the figure is confusing and should be improved.

Thank you for this comment. We have redesigned Figure 1.

Line 446. As far as I know, the antioxidant role of apoA-II has not been demonstrated, I suggest delete apoA-II from this sentence.

We agree. We have deleted apoA-II from the sentence (Line 446)